# Sclerosing Microcystic Adenocarcinoma Arising from the Tongue: A Case Report and Literature Review

**DOI:** 10.3390/diagnostics12051288

**Published:** 2022-05-21

**Authors:** Yi-Ying Lee, Tzer-Zen Hwang, Ying-Tai Jin, Chien-Chin Chen

**Affiliations:** 1Department of Pathology, E-DA Hospital, I-Shou University, Kaohsiung 82445, Taiwan; yiying122@gmail.com; 2Department of Otolaryngology-Head and Neck Surgery, E-DA Hospital, I-Shou University, Kaohsiung 82445, Taiwan; 3School of Medicine, College of Medicine, I-Shou University, Kaohsiung 82445, Taiwan; 4Department of Pathology, Taiwan Adventist Hospital, Taipei 10556, Taiwan; yingtai@tahsda.org.tw; 5Department of Pathology, National Cheng Kung University Hospital, Tainan 704, Taiwan; 6Department of Pathology, Ditmanson Medical Foundation Chia-Yi Christian Hospital, Chiayi 600, Taiwan; 7Department of Cosmetic Science, Chia Nan University of Pharmacy and Science, Tainan 717, Taiwan; 8Department of Biotechnology and Bioindustry Sciences, College of Bioscience and Biotechnology, National Cheng Kung University, Tainan 701, Taiwan

**Keywords:** adenocarcinoma, head and neck, microcystic adnexal carcinoma, mucosa, salivary gland, sclerosing microcystic adenocarcinoma, squamous cell carcinoma

## Abstract

Sclerosing microcystic adenocarcinoma is a rare and recently characterized cancer that affects the mucosal surfaces of the head and neck without adnexal involvement. Histologically, microcystic adnexal carcinoma of the skin resembles it. It does, however, contain unique characteristics that merit our attention for potential diagnostic errors. Therefore, we present a 48-year-old male with sclerosing microcystic adenocarcinoma of the tongue, along with a full discussion and a brief review of pertinent literature.

## 1. Introduction

Sclerosing microcystic adenocarcinoma (SMA) is a rare type of cancer that affects the mucosal surfaces of the head and neck [1,2,3,4,5,6,7,8]. Histologically, the tumor resembles cutaneous microcystic adnexal carcinoma (MAC) but shows no evidence of adnexal connections, characterized by thin nests and cords of squamous and basaloid cells forming ductal structures in the desmoplastic stroma [1,2,3,4,5,6,7,8]. Both SMA and MAC have evident local infiltration and perineural invasion [1,2,3,4,5,6,7,8,9]. The term SMA was first coined by Mills et al. in 2016 [1]. It has not been introduced as a distinct type in the previous edition of tumor classification of the World Health Organization (WHO, Geneva, Switzerland) in 2017 [10] but has been recently recognized as a new entity in the newest edition [11]. In total, 13 cases of SMA have been reported in the literature; 11 concerned the oral cavity, including the floor of the mouth and the tongue, one concerned the nasopharynx, and another concerned the parotid gland [1,2,3,4,5,6,7,8]. Performing an accurate histological diagnosis of the tumor is difficult, especially if a small incisional biopsy is made and an evident infiltrative pattern is lacking [8]. Herein, we report an additional case and provide a brief literature review to raise awareness about this rare tumor.

## 2. Case Presentation

A nonsmoking man, aged 48 years, who did not chew betel nut, had a history of right tongue squamous cell carcinoma (SCC) and had received partial glossectomy 9 years previously at a medical center (Hospital A). He presented to Hospital A with dysarthria that had persisted for 6 months. A physical examination and initial head and neck computed tomography (CT) scan indicated no obvious tumor-like mass. Because the man’s symptoms persisted, head and neck magnetic resonance imaging (MRI) was arranged 4 months later and revealed a left tongue tumor. A biopsy of the tongue tumor was performed, and the pathology report revealed adenocarcinoma, intermediate-grade. Therefore, the patient visited our hospital for further survey. A physical examination revealed a 4 × 4 cm^2^ left ventral tongue tumor. The 18F-FDG positron emission tomography (PET)/CT revealed a hypermetabolic process in the tongue, consistent with recurrent tongue cancer without the involvement of regional lymph nodes or distant metastases (Figure 1A–D). The patient subsequently underwent wide excision of the left tongue tumor with bilateral selective neck dissection.

In the gross examination, an infiltrative ulcerated tumor was identified at the left lateral tongue border (Figure 1E); the tumor, measuring 5 × 4 cm^2^, crossed the midline and involved the ipsilateral tongue base and part of the extrinsic tongue musculature. Multiple small lymph nodes were noted at the bilateral neck and measured up to 0.9 × 0.8 cm^2^. Microscopically, the tumor exhibited diffusely infiltrating cords, tubules, and strands of epithelial cells with biphasic components of ductal and myoepithelial cells (Figure 2A,B). Perineural invasion and wide infiltration into adjacent skeletal muscle were observed (Figure 2C,D). In immunohistochemistry, CK7 and p40 highlighted a dual cell population of luminal and abluminal cells (Figure 2D,E). The neoplastic cells were negative for CD117, and the proliferation index, as determined by Ki-67, was low (<5%). No dysplasia was noted in the overlying mucosa. The surgical margin at the base of the tongue had tumor involvement. All dissected lymph nodes were negative for malignancy. In terms of histomorphologic findings and location, the tumor most closely resembled the SMA of the tongue. After discussing with the radiation oncologist for postoperative radiotherapy, the patient declined the radiotherapy for fear of side effects but received complete adjuvant chemotherapy due to margin positivity. Follow-up head and neck magnetic resonance imaging revealed no evidence of recurrence 23 months after surgery. The patient is in good health 2 years and 6 months after surgery and adjuvant chemotherapy.

## 3. Discussion

SMA is a recently identified uncommon type of cancer with a similar manifestation to skin MAC [1,11]. Mills et al. first reported and named the type of cancer [1]. Reports indicate that SMA affects intraoral or other mucosae of the head and neck. In MAC and SMA, thin nests and cords of basaloid cells form cystic spaces and ducts within sclerotic stroma [1,6,8]. Substantial infiltrative growth and perineural invasion are noted characteristics of MAC and SMA [1,9]. Generally, MAC occurs in head and neck skin and is assumed to originate from the pilosebaceous-apocrine unit. Although it has a similar histomorphology to MAC, SMA may originate in minor salivary glands embedded in nonadnexal mucosal sites of the head and neck [1,11]. In SMA, no evidence of skin adnexal tumors has been identified [1,3,4].

Table 1 summarizes the clinicopathological characteristics of reported cases and the case described in this report [1,2,3,4,5,6,7,8]. In total, 14 cases were reviewed; a female preponderance (9:5) was noted, and the median age of affected patients was 55.9 years (age range: 41 to 73 years). The primary sites were the tongue (n = 9), floor of the mouth (n = 3), nasopharynx/ clivus (n = 1), and parotid gland (n = 1). A dual cell population was identified through immunohistochemistry, with luminal cells demonstrating positivity for low- and high-molecular-weight cytokeratins and surrounding myoepithelial cells exhibiting immunoreactivity against smooth muscle actin and S100. CD117 staining (c-kit) was negative [8]. An immunosuppressed status might play a role in the pathogenesis of SMA [1]. Mills et al. indicated that two affected patients had previously received systemic chemotherapy and radiotherapy. Zhang et al. revealed that one affected patient was diagnosed with multiple sclerosis and was receiving immunosuppression therapy. Jiang et al. stated that one affected patient had psoriatic arthritis and received immunosuppression therapy. Tan et al. revealed that one affected patient who had SMA in the parotid gland had a history of nasopharyngeal carcinoma and was undergoing radiotherapy [1,5,6,8]. Interestingly, a dominant proportion of the published cases and our case had a history of various neoplasms, such as adenoid cystic carcinoma (AdCC), acute myeloid leukemia, ovarian benign mucin-ous cystadenoma, nasopharyngeal carcinoma, and SCC, and/or immune-related diseases, including mesangiocapillary glomerulonephritis, multiple sclerosis, and psoriatic arthritis.

Unlike MAC of the skin, which often culminates in local recurrence and uncommon lymph node metastases, SMA appears to have an indolent clinical manifestation without distant metastasis or nodal involvement. All patients were still living after a median follow-up duration of 21 months (range: 4–60 months) without distant and local recurrence [1,2,3,4,5,6,7,8]. Jiang et al. reported that one patient’s disease had the molecular and genetic characteristics of SMA [6]. Their study is the only one that provides details of molecular changes in SMA with loss of function mutations in CDK11B [6]. Notably, these researchers identified no association between SMA and known MAC mutations [6].

Performing a histological diagnosis of this rare disease is difficult, especially when small biopsy specimens are obtained in which diagnostic characteristics are not apparent. For example, the basophilic desmoplastic stroma surrounding the neoplastic ducts can be mistaken for the atrophic and sclerotic lobules of salivary gland tissue when the infiltrating pattern is not biopsied or only present focally; the impression of a benign lesion may be obtained. Additionally, the paucicellular stroma could be regarded as reactive changes, especially when biopsies are repeatedly performed. Therefore, an inconclusive diagnosis may be obtained after a small incisional biopsy [8]. A diagnosis can only be made if, in the resected specimen, solid evidence exists of wide infiltration into adjacent structures, such as skeletal muscle and unequivocal perineural invasion [8]. In addition, assessments of margin status are extremely challenging, especially during intraoperative consultation, because infiltrative growth patterns and the paucicellular nature of the tumor might not be observed. In our patient, an intraoperative margin assessment seemed to reveal negative findings; however, for the deep/base margin, positive findings were obtained through a comparison of that tumor with the main tumor specimen. Pathologists should have a greater awareness of this issue during intraoperative margin assessments. Diagnosis relies heavily on careful morphological evaluations in large surgical specimens.

For SMA, differential diagnoses involve eliminating malignant neoplasms such as AdCC, SCC, low-grade mucoepidermoid carcinoma (MEC), polymorphous adenocarcinoma (PLGA), mammary analog secretory carcinoma, and specific benign lesions such as sclerosing sialometaplasia and chronic sclerosing sialadenitis [1,5]. The most critical step in a differential diagnosis is distinguishing SMA from SCC; SCC is the most similar malignancy in differential diagnoses related to the head and neck and requires a substantially different management approach. Unlike SCC, SMA lacks marked cytologic atypia and surface epithelial dysplasia. Although the relevant infiltrative tumor nests have an SCC-like appearance, dyskeratosis and a keratin pearl formation are lacking, and the presence of bilayer cell structures with a relatively bland appearance suggests that SCC can be ruled out. Our patient reported a history of SCC on the right tongue nine years previously. However, no histological slides were available for a pathological review. This reflected the difficulty of diagnosing SCC. Despite locally aggressive histological features, including infiltrative borders and the perineural invasion of SMA, the prognosis is favorable; pathologists should pay close attention to symptoms and signs to avoid misdiagnosis.

AdCC, specifically the tubular pattern, may have histomorphological findings that overlap with SMA’s, but basophilic secretion and angulated hyperchromatic nuclei not observed in SMA are observed in AdCC. Moreover, SMA lacks a cribriform growth pattern. CD117 (c-Kit) immunostaining, which is usually positive in AdCC but negative in SMA, could be helpful for a precise diagnosis [5]. The PLGA border is relatively circumscribed at low power magnification with high cellularity. By contrast, SMA exhibits diffuse infiltration and lower cellularity. The presence of dense sclerosing stroma and the absence of architectural variability are valuable features for differentiating SMA from PLGA. Secretory carcinoma is characteristic of intraluminal eosinophilic secretions; however, the substantial sclerotic stroma is atypical in mammary analog secretory carcinoma patients. Often, low-grade MEC is cystic and can be eliminated in differential diagnoses based on the presence of an admixture of intermediate cells, epidermoid cells, and scattered mucocytes or nests of mucocytes in varying proportions. Furthermore, low-grade MEC lacks the characteristic sclerotic/hyalinized stroma observed in SMA. One exception is an uncommon eosinophilic variant of MEC with areas of hyalinization [2]. However, this rare variant manifests in major salivary glands, mainly the parotid gland, and prominent eosinophilia is not observed in SMA.

PLGA exhibits architectural diversity; it can manifest as uniform neoplastic cells arranged in streaming cords, tubules, and cribriform patterns [10]. During microscopic examinations, infiltrative single cells at the tumor periphery and perineural invasion are frequently encountered. PLGA must be differentiated from SMA, especially the low-grade variant. PLGA (even the low-grade variant) exhibits substantially higher architectural variability in microscopic examinations than SMA; it can be characterized by forming fascicles, tubules, and small, solid, and papillary areas. The most typical diagnostic features are concentric layers or whirl-like structures of neoplastic cells around the central nidus. No dense, sclerotic stroma or bilayered ducts of SMA are observed in PLGA. Although both tumors exhibit infiltrative growth at the periphery and relatively cytologic uniformity, PLGA manifests as cellular, circumscribed tumor nests, whereas SMA is widely dispersed and infiltrative within a paucicellular stroma. Immunohistochemical studies would be helpful for a differential diagnosis. PLGA stains S-100 protein uniformly, which is not the case with SMA. According to the most recent WHO tumor classification, adenocarcinoma, not otherwise specified (NOS), should be considered during diagnoses. However, the term represents a diagnosis by exclusion, and SMA exhibits characteristic morphological features, as observed by Mills et al. [1]. Several known translocation-associated molecular features have been identified in differential diagnoses, including adenoid cystic carcinoma (MYB-NFID), low-grade MEC (MECT/MAML), and mammary analog secretory carcinoma (ETV6/NTRK3) [1,6,10]. In their molecular study of SMA, Jiang et al. identified loss of function mutations in CDK11B [6]. Thus, tests for revealing molecular changes might be of value for differentiating these morphologically low-grade tumors from SMA.

Sclerosing sialometaplasia is a benign reactive lesion that should be differentiated from SMA. Helpful diagnostic features that point toward a malignancy—specifically, SMA—are diffusely infiltrative growths beyond the extent of minor salivary glands and the presence of marked perineural invasion [5]. A lobular configuration without perineural invasion is observed in sialometaplasia [5]. Related studies have addressed the difficulties in identifying infiltrative histomorphological features on initial biopsies and during intraoperative evaluations. Paucicellular structures with ample hyalinized stroma are the most problematic histomorphological features [5,8]. It is easy to mistake them for atrophic features that indicate the presence of chronic sialadenitis [5]. In our patient, a constellation of malignant features was only observed in subsequently resected specimens in which widely infiltrative growth, including extensive perineural invasion, was noted.

Given the rarity of this diagnosis with the availability of only a few published case reports, choosing an optimal therapy may be difficult; such choices could follow treatment practices for MAC [1]. Tumor wide excision with safe margins may be essential for treating cutaneous MAC on the lip, but for SMA in oral mucosa beside the lip, a less radical surgical approach may be suitable when no recurrent diseases are reported; however, such approaches must be validated by evidence from further case reports or series [2].

## 4. Conclusions

SMA is a recently identified malignant neoplasm affecting the head and neck; it has indolent clinical behavior and a favorable prognosis despite infiltrative growth. Pathologists should be aware of this rare entity when encountering a tumor with neoplastic cells exhibiting low-grade cytological atypia, a dual cell differentiation (luminal and myoepithelial), and the formation of small tubules, nests, microcysts, and/or ducts. We present a further case report and thorough literature review to improve the understanding of this extraordinarily rare type of cancer.

## Figures and Tables

**Figure 1 diagnostics-12-01288-f001:**
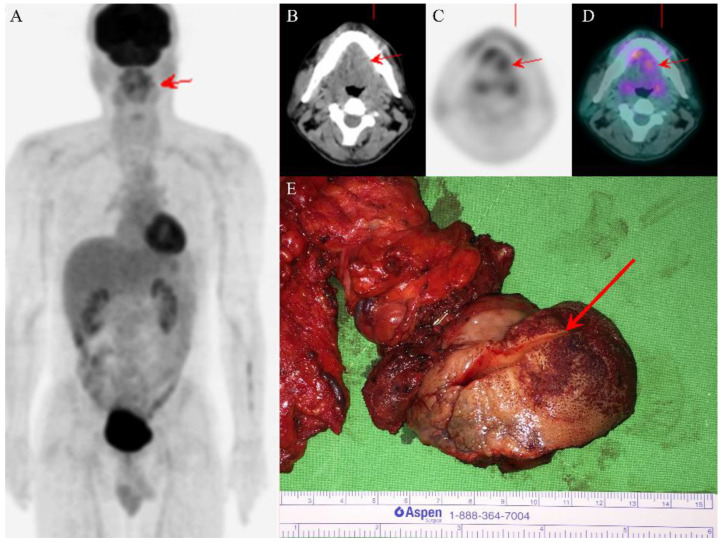
PET/CT images revealed one ill-defined infiltrative tumor (red arrows) over the left tongue with high uptake of fluorodeoxyglucose and absence of regional lymph node involvement or distant metastasis (**A**–**D**). The wide excision of the left tongue tumor grossly showed ulcerative mucosa and one ill-defined white submucosal tumor after cutting (red arrow) (**E**).

**Figure 2 diagnostics-12-01288-f002:**
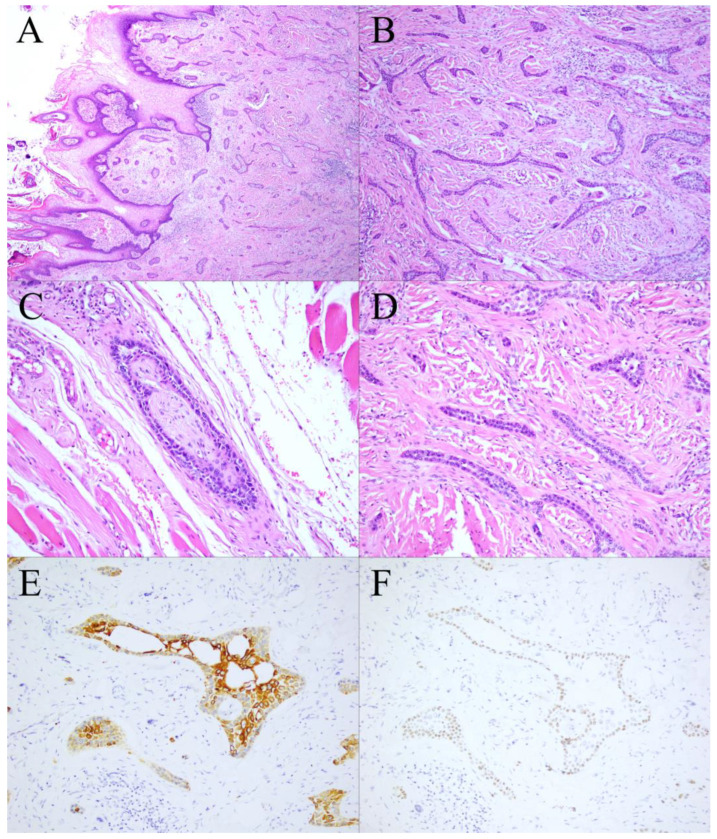
The tumor exhibited diffusely infiltrating cords, tubules, and strands of epithelial cells without overlying mucosal dysplasia ((**A**,**B**), ×40). Higher magnification showed wide infiltration into adjacent skeletal muscle and perineural invasion ((**C**), ×200). Bilayer cell structures with a relatively bland appearance were noted ((**D**), ×200). In immunohistochemistry, CK7 and p40 highlighted a dual cell population of luminal and abluminal cells ((**E**): CK7, ×200; (**F**): p40, ×200).

**Table 1 diagnostics-12-01288-t001:** A summary of case reports of sclerosing microcystic adenocarcinoma with clinicopathologic features.

Case	Author [Reference]	Age	Sex	Location	Immunohistochemical Results	Treatment/Follow-Up (Months)	Note
1	Mills et al. [1]	41	F	Base of tongue	Negative: CD117/c-kit	Not stated/not stated	History of soft palate adenoid cystic carcinoma s/p radiotherapy 7 years prior; no lymph node involvement
2	Mills et al. [1]	47	F	Anterior tongue	Not available	Not stated/not stated	No lymph node involvement
3	Mills et al. [1]	73	M	Nasopharynx and clivus	Positive: CK cocktail, BerEP4, S100+ myoepithelial cells. Negative: CD117/c-kit, CK20, CEA	Not stated/not stated	No lymph node involvement
4	Mills et al. [1]	54	F	Floor of mouth	Positive: CK cocktail, SMA+ myoepithelial cells.	Not stated/not stated	No lymph node involvement
5	Mills et al. [1]	48	F	Floor of mouth	Positive: CK5/6 Negative: CK7 Proliferation index (Ki-67): 5%	Chemoradiotherapy/ disease free (24)	History of AML s/p stem cell transplantation 18 years prior; no lymph node involvement
6	Wood et al. [2]	68	F	Tongue tip	Positive: CK7, CAM5.2, p63, S100 Negative: CK20, ER, PR, TTF-1, CDX2, SMA, Calponin, CD117/c-kit Proliferation index (Ki-67): <5%	Excised with clear margins/disease free (60)	No distant metastasis
7	Wood et al. [2]	49	F	Right lateral tongue	Positive: CK7, p63, S100, CK5/6 Negative: CK20, ER, PR, TTF-1, CDX2, SMA, Calponin Proliferation index (Ki-67): <5%	Excised with clear margins/disease free (14)	History of mesangiocapillary glomerulonephritis and left ovarian benign mucinous cystadenoma; no lymph node involvement; no distant metastasis
8	Petersson et al. [3]	70	F	Left posterior tongue	Positive: CK7, LMWCK, BerEP4, HMWCK, CK5, CK18, p63, PASD Negative: CK20, TTF-1, SMA heavy chain, SMA, bcl-2, p53, p21 Proliferation index (Ki-67): 2–4%	Excised with involved margins and adjuvant radiotherapy/disease free (21)	No lymph node involvement; no distant metastasis
9	Schipper et al. [4]	65	M	Tongue	Positive: CAM5.2, CEA (inner layer), EMA	Radiotherapy/no change in tumor size (21)	No distant metastasis
10	Zhang et al. [5]	55	F	Floor of mouth	Positive: AE1/3, p63, CK5/6, CK7, EMA, p63, p40, S100 Negative: androgen receptor, SOX10, CD34, beta-catenin, CD68, IgG, IgG4, CD117/c-kit	Excised with involved margins and adjuvant radiotherapy/disease free (10)	Multiple sclerosis and a family history of *BRCA* gene mutation; no distant metastasis
11	Jiang et al. [6]	41	F	Right tongue tip	Positive: CK7 (inner layer), p40 and p63 (outer layer) Negative: SOX10, CD117/c-kit, S100, mammoglobin, GATA3	Excised with clear margins/not stated	History of psoriatic arthritis with immunosuppressive therapy; no lymph node involvement
12	Zhang et al. [7]	51	M	Left tongue	Positive: CK5/6, CK8/18, EMA, CK7, p63, S100, CD10, SMA Negative: CK20, calponin, myb, her2, bcl2, p53, CD117/c-kit, cd43 Proliferation index (Ki-67): <5%	Excision/disease free (38)	No lymph node involvement; no distant metastasis
13	Tan et al. [8]	73	M	Left parotid gland	Positive: EMA, CK7, SOX10, p63, S100, PAS (for eosinophilic secretions) Negative: CD117 Proliferation index (Ki-67): 5%	Excised with involved deep resection margin/disease free (4)	History of nasopharyngeal carcinoma s/p radiotherapy 23 years prior; concurrent tonsillar SCC; no lymph node involvement
14	Current case	48	F	Left tongue	Positive: CK7, p40, p63 Negative: CD117/c-kit Proliferation index (Ki-67): <5%	Excised with involved margins/disease free (30)	History of right tongue SCC s/p partial glossectomy 9 years prior; no lymph node involvement; no distant metastasis

AML, acute myeloid leukemia; BerEp4, Ep-CAM/epithelial specific antigen; CEA, carcinoembryonic antigen; CK, cytokeratin; EMA, epithelial membrane antigen; ER, estrogen receptor; HMWCK, high molecular weight cytokeratin; LMWCK, low molecular weight cytokeratin; MYB, v-Myb avian MYB viral oncogene homolog; PR, progesterone receptor; SCC, squamous cell carcinoma; SMA, smooth muscle actin; SOX10, SRY-related HMG-box 10.

## Data Availability

Not applicable.

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
