# Peer review of "Sclerosing Microcystic Adenocarcinoma Arising from the Tongue: A Case Report and Literature Review"

_diagnostics, 2022, doi:10.3390/diagnostics12051288_

Round 1
Reviewer 1 Report
Congratulations to the authors for reporting a rare entity which would add to the existing literature.
I would suggest modifying the write up of abstract especially the introduction which has very close resemblance to the report by Zhang R et al.( 2019) One query is, did the patient receive adjuvant radiotherapy or only chemotherapy and if so why only chemotherapy. If we review all the cases what is very striking is of the 14 cases 5 had previous history of cancer and 2 had other diseases related to immune system.
Author Response
The authors are very grateful to the reviewers for their constructive comments on the paper. We have thought through each in-depth and provided a point-by-point response following the reviewers’ comments below.
Reviewer 1
Comments:
Congratulations to the authors for reporting a rare entity which would add to the existing literature.
I would suggest modifying the write up of abstract especially the introduction which has very close resemblance to the report by Zhang R et al.( 2019)
- Thank you for your recommendation. We have rewritten our abstract and introduction according to your advice. The revised parts were highlighted in red. We are grateful for your sincere comments.
One query is, did the patient receive adjuvant radiotherapy or only chemotherapy and if so why only chemotherapy.
- The patient had only postoperative chemotherapy because he worried about the side effects of radiotherapy after discussion with radiation oncologists. Therefore, we have revised the description of postoperative therapy in the last paragraph of the case history to make it clear. The revised parts were highlighted in red. We are grateful for your sincere comments.
If we review all the cases what is very striking is of the 14 cases 5 had previous history of cancer and 2 had other diseases related to immune system.
- Thank you for your sincere comments. It’s true that a dominant proportion of the published cases and our case had a history of various neoplasms (soft palate adenoid cystic carcinoma, acute myeloid leukemia, ovarian benign mucin-ous cystadenoma, nasopharyngeal carcinoma, and squamous cell carcinoma) and/or immune-related diseases (mesangiocapillary glomerulonephritis, multiple sclerosis, and psoriatic arthritis). Since the types of tumors and immune diseases have no common pathways and characteristics, large multicenter cohorts and large-scaled molecular studies are essential for further characterization and investigation.

Reviewer 2 Report
interesting case report , rare disease in the literature
Author Response
The authors are very grateful to the reviewers for their constructive comments on the paper. We have thought through each in-depth and provided a point-by-point response following the reviewers’ comments below.
Reviewer 2
Comments:
interesting case report, rare disease in the literature.
Response to comments
- We appreciate your professional comments.

Reviewer 3 Report
As you have mentioned that Sclerosing microcystic adenocarcinoma was recently described in 2016 hence the number of cases would be less, But actual frequency is still unknown,
You mentioned that the post operative margin were positive after Wide local excision and patient was put on chemotherapy treatment. But as a routine such patients are managed by re surgery or concurrent chemoradiotherapy.
Author Response
The authors are very grateful to the reviewers for their constructive comments on the paper. We have thought through each in-depth and provided a point-by-point response following the reviewers’ comments below.
Reviewer 3
Comments:
#1 As you have mentioned that Sclerosing microcystic adenocarcinoma was recently described in 2016 hence the number of cases would be less, But actual frequency is still unknown.
- Really appreciate your precise comments. The entity is new and uncommon, so its incidence needs more cases’ publishment and further investigation.
#2 You mentioned that the post operative margin were positive after Wide local excision, and patient was put on chemotherapy treatment. But as a routine such patients are managed by re surgery or concurrent chemoradiotherapy.
- Thank you for the sincere comments. Honestly, we arranged postoperative concurrent chemoradiotherapy for him, but he refused to receive the radiotherapy due to the fear of the side effects. Therefore, he eventually received only complete chemotherapy after the operation. Luckily, he does not have tumor recurrence till now. To make the reason straightforward, we have revised the description of postoperative therapy in the last paragraph of the case history. The revised parts were highlighted in red.

Round 2
Reviewer 3 Report
Improved